# Chemical Conversations

**DOI:** 10.3390/molecules30030431

**Published:** 2025-01-21

**Authors:** Jana Michailidu, Olga Maťátková, Alena Čejková, Jan Masák

**Affiliations:** Department of Biotechnology, University of Chemistry and Technology Prague, Technicka 5, 166 28 Prague, Czech Republic; jana.michailidu@vscht.cz (J.M.); olga.matatkova@vscht.cz (O.M.); alena.cejkova@vscht.cz (A.Č.)

**Keywords:** microorganisms, plants, animals, interspecies communication, signaling molecules, chemical conversation

## Abstract

Among living organisms, higher animals primarily use a combination of vocal and non-verbal cues for communication. In other species, however, chemical signaling holds a central role. The chemical and biological activity of the molecules produced by the organisms themselves and the existence of receptors/targeting sites that allow recognition of such molecules leads to various forms of responses by the producer and recipient organisms and is a fundamental principle of such communication. Chemical language can be used to coordinate processes within one species or between species. Chemical signals are thus information for other organisms, potentially inducing modification of their behavior. Additionally, this conversation is influenced by the external environment in which organisms are found. This review presents examples of chemical communication among microorganisms, between microorganisms and plants, and between microorganisms and animals. The mechanisms and physiological importance of this communication are described. Chemical interactions can be both cooperative and antagonistic. Microbial chemical signals usually ensure the formation of the most advantageous population phenotype or the disadvantage of a competitive species in the environment. Between microorganisms and plants, we find symbiotic (e.g., in the root system) and parasitic relationships. Similarly, mutually beneficial relationships are established between microorganisms and animals (e.g., gastrointestinal tract), but microorganisms also invade and disrupt the immune and nervous systems of animals.

## 1. Introduction

A fundamental characteristic of living organisms is their capacity and need to communicate with one another. Organisms use a variety of means to communicate, and at the top of the list, we could place verbal communication in humans in the form of spoken language or written text. In addition to other forms of acoustic means, we encounter mechanical communication, optical communication, and one of the oldest types—chemical communication, which is the focus of this review. In many cases, this type of communication involves relatively complex relationships between the participants and resembles a conversation in the form of speech.

The chemical language that enables communication between organisms in nature is perhaps the most extensive communication system, operating at multiple levels and creating complex relationships. It is, therefore, not possible to give a comprehensive picture of all these aspects of the communication process in the following review. The main focus will be on prokaryotic and eukaryotic unicellular organisms—both among them and in their relationship to higher organisms. In the microbial world research on chemical communication has shown that these organisms can act in a coordinated manner, and we can actually speak about a form of multicellularity [1,2]. The next part of the text is devoted to the communication of these microorganisms with higher organisms and the resulting positive or negative relationships that develop between them. The variety of specific relationships has necessarily led to a limited selection of perhaps interesting examples.

## 2. Chemical Languages of Microorganisms

### 2.1. Communication Between Cells Belonging to the Same Population

It has been known for many years that the behavior of microbial populations can have aspects of the behavior of multicellular organisms. In this context, a communication mechanism called quorum sensing has been well studied; it is the regulation of gene expression in response to changes in cell density [3,4]. Quorum sensing relies on the extracellular production of autoregulatory (autoinducer, AI) molecules (see Table 1), with their production typically activated by external stimuli. Typically, as the number of cells increases, the concentration of AI molecules increases. Once a sufficient cell density (quorum) is reached, AI molecules bind to transmembrane or cytoplasmic receptors, initiating a cascade of intracellular reactions. This process often involves two-component signal transduction systems and can lead to phenotypic changes across the entire population, triggering a coordinated response. At a threshold concentration, autoinducers can interact with multiple targets, inducing a metabolic switch [5]. These processes include symbiosis, virulence, competence, conjugation, antibiotic production, motility, sporulation, biofilm formation, etc. [4]. There has also been a finding that, in some cases, the individual phenotypic changes may be dependent on each other.

This chemical communication mechanism is used by G^+^ and G^−^ bacteria, as well as unicellular eukaryotes (yeasts and micromycetes). In Gram-negative bacteria, autoinducer molecules are often acyl-homoserine lactones (AHLs), which can freely diffuse into the cell and interact with cytoplasmic receptors, such as LuxR-type proteins. In contrast, Gram-positive bacteria utilize short, modified peptides as AIs that bind to transmembrane histidine kinase receptors, initiating a two-component signal transduction system. Archaea employ similar autoregulatory molecules for quorum sensing, including phenylethanol, tryptophol, and tyrosol [6]. Table 1 and Figure 1 provide examples of the most common signaling molecules used by microorganisms for communication within a cell population [7,8,9,10].

Microorganisms have receptors that recognize signaling molecules produced by other microbial species. The fact that different microbial species produce regulatory molecules with a similar chemical structure also plays a role in this. Microbial responses to ’foreign’ signaling molecules have become more complex over time [11,12]. We could also talk about a kind of communication noise. This refers to the interference or disruption in microbial signaling pathways caused by environmental factors or the presence of competing microorganisms. Similarly, intense communication takes place not only between prokaryotes—bacteria, archaea—but also between bacteria and unicellular eukaryotes (micromycetes, etc.) [13].

### 2.2. Chemical Communication of Competitors and Predators

Microorganisms produce substantial quantities of chemical compounds, typically secondary metabolites, that influence other organisms in their environment, which are competing for the same resources or acting as predators (see Figure 1). This form of interaction can resemble chemical warfare. A form of deterrence is the least aggressive method. More than 80 years ago, Singh [14] described that amoebae avoid microorganisms such as *Serratia marcescens* or *Chromobacterium violaceum*, which produce toxic pigments (prodigiosin and violacein, respectively) while feeding on non-pigmented strains. Similarly, the beetle *Carpophilus hemipterus* avoids the sclerotia of *Aspergillus flavus*, which contains toxic secondary metabolites but feeds on other parts of the fungus [15]. Microorganisms generally compete with higher animals for the same nutrient sources (agricultural products, food, etc.). They can be a deterrent to animals through visual changes (pigmentation) or the formation of odor compounds [16]. Microbial production of various toxins is also involved in food spoilage. If the animal does not reject the spoiled food based on prior information (color, odor), it will be affected by the toxin, which can lead to its death, which is again advantageous for the microorganism [17].

A very well-known large group of substances are siderophores (see Figure 2), compounds produced by microorganisms that are capable of sequestering an important nutrient in the environment—iron. A lack of iron for other microorganisms can severely limit their growth. Thus, they are molecules that indirectly affect (modulate) the growth of their competitors [18,19]. Siderophores are currently considered to be signaling molecules, highlighting their significant role in bacterial cell communication [20].

Microorganisms produce a variety of chemical compounds which are designed to destroy a member of a particular community. The production of these substances is regulated by quorum sensing and depends on the concentration of autoinducers at a given phase of the cell cycle. Antibiotics, toxins that only act against a limited group of microorganisms, are used for this type of relationship between microorganisms (see Figure 2). Due to the high specificity of their action at certain target sites, the target populations develop defense or resistance mechanisms. Conversely, at low, sub-inhibitory concentrations, antibiotics can act as signaling molecules to enhance defense responses in single or multi-species microbial communities, for example, by inducing biofilm formation [21]. Some bacteria have another lethal communication system known as the type VI secretion system. This allows them to inject toxic proteins directly into the cells of microbial rivals [22].

One example of unilaterally beneficial chemical communication between microorganisms and plants is the interference with the plant hormone system. Gibberellins are plant hormones that, together with other substances such as auxins, cytokines, etc., regulate a number of plant processes (see Figure 2). *Fusarium moniliforme* is a plant pathogen that can produce gibberellins, often in significant quantities, after infecting a plant and significantly affecting the development of the plant and causing its unhealthy development (e.g., bakanae rice). The damaged plant then provides an ideal environment and nutrient source for the fungus’s life cycle [23].

Microorganisms produce potent toxins that kill animals as competitors for nutrients or as an important source of nutrients. These toxins are designed to target either major organs (e.g., liver) or central systems (nervous, cardiovascular, immune, etc.).

For example, the diverse range of bacterial neurotoxins includes botulinum toxins, tetanus neurotoxin, and various enterotoxins [24]. The production and transport of Streptococcus pneumoniae toxins are one of the major causes of severe brain cell damage [25]. An example of a cardiotoxin is the thermo-labile enterotoxin B produced by enterotoxigenic *Escherichia coli*, which can disrupt critical host functions [26].

The relationships between fungi and insects offer countless possible types of communication, from symbiotic behavior to predation. An intriguing example of fungal predation is observed with Ophiocordyceps, which can exert control over the behavior of ants following spore infection. Once infected, the ant exhibits ‘zombie-like’ behavior, eventually leading to death. This manipulation is facilitated by bioactive compounds produced by the fungus. Recently, the neurotoxin aflatrem was identified as a contributing factor, slowing ant movement and causing unsteady, staggering behavior [27,28].

The pathogenic fungus *Cordyceps militaris* infects butterfly larvae through a complex interaction. It produces 3′-deoxyadenosine, known as cordycepin, which exhibits antimicrobial properties and suppresses the larvae’s immune response by downregulating specific immune-related genes. This suppression allows the fungus to thrive within the host. Interestingly, the concentration of cordycepin is regulated to control bacterial infections within the larvae, ensuring the host remains alive long enough for the fungus to develop sufficient biomass. This growth culminates in the formation of spore-producing structures that emerge from the deceased host, facilitating the release and dissemination of spores [29].

Another example is the bacterium *Xenorhabdus nematophila*, which has a mutualistic relationship with a certain species of nematodes (Heterorhabditidae) and is also a pathogen of insects. The bacterium–nematode complex initiates infection by entering the insect host. Once inside, the nematodes release their symbiotic bacteria into the insect’s hemocoel. These bacteria produce toxins that rapidly kill the host. Subsequently, the nematodes progress through four developmental stages within the insect cadaver. During this period, the bacteria recolonize the nematodes, which eventually emerge from the cadaver to seek new hosts [30].

Micromycetes produce a wide range of structurally diverse toxic secondary metabolites collectively known as mycotoxins. These compounds include several major classes, such as aflatoxins (produced by *Aspergillus* and known for their carcinogenic properties), trichothecenes (produced by, e.g., *Fusarium* species, causing immunosuppression and cytotoxic effects), ochratoxins (known for nephrotoxicity and produced by *Aspergillus* and *Penicillium* species), fumonisins (linked to neural and hepatic toxicity, primarily produced by *Fusarium* species), patulin (a neurotoxic compound produced by *Penicillium* and *Aspergillus* species), and ergot alkaloids (produced by *Claviceps* species, causing vasoconstriction and neurological symptoms). Their effects encompass a multitude of other organisms, including animals, which are complex and can damage individual organs or modulate the activity of central systems [31].

### 2.3. Cooperative Chemical Signaling

Chemical communication between microorganisms or with higher organisms does not always have to be confrontational. It is quite common for chemical signaling to induce and sustain mutually beneficial relationships.

In infectious diseases, pathogenic microorganisms can form mutually beneficial communities in the form of polymicrobial biofilms. For example, *Pseudomonas aeruginosa* is able to coexist in biofilms with a variety of bacteria, fungi, and viruses (e.g., *Burkholderia cepacia*, *Staphylococcus aureus*, *Enterococcus faecalis*, *Streptococcus* spp., *Acinetobacter baumannii*, *Actinomyces* spp., *Propionibacterium* spp., *Aspergillus fumigatus*, *Candida albicans*, respiratory syncytial virus (RSV), human rhinovirus, influenza virus, and others). However, these synergistic relationships are very fragile and can easily be switched to a competitive form by environmental changes such as the production of regulatory/signaling molecules [32].

There are very complex consortia in the soil that communicate/cooperate with each other very intensively. Microorganisms thus often form complex metabolic systems that provide the necessary nutrients to the populations present [33]. The relationships between legume root systems and rhizobia in the rhizosphere are a well-known example of this phenomenon. The transition from the saprophytic phase of rhizobia to the symbiotic phase is induced by the action of chemoattractants contained in plant exudates. The rhizobia become part of the nodule. Nodule formation is initiated by the exchange of compatible signals between rhizobia and legumes. Rhizobia enter the root tissue and, after further transformation, fix nitrogen [34]. Another example of plant–microbe interactions are signaling molecules such as strigolactones released by plant roots, which attract beneficial arbuscular mycorrhizal fungi, fostering symbiotic relationships that enhance nutrient uptake.

Mutually beneficial relationships between microbial endophytes and their hosts are very common. Endophytes strive to maintain a healthy host environment. For instance, the endophyte *Acremonium lolii*, found in Lolium grass, produces two alkaloids. The first, lolitrem B, can induce neurological issues in cattle, discouraging grazing. The second, peramine, acts as a natural insect repellent [35,36].

The chemical communication between microorganisms and animals is known. However, there is a lack of understanding of the complexity of the production of molecules that are beneficial to animals and also to microorganisms. Information on the production of chemicals by microorganisms (inhabiting mammal scent glands) and their use by animals in olfactory communication is gradually increasing [37]. A very interesting example is the study by Theis et al. Using advanced instrumental analytical techniques, they found a high correlation between bacterial communities in hyena scent gland secretions and odor profiles. They found significant differences in the bacterial communities and odor profiles of two hyena species. The bacterial communities and odor profiles also differed between individual hyena clan members—males, pregnant females, and lactating females [38]. Not to be overlooked is the symbiotic relationship between the marine bacterium Vibrio fischeri and the Hawaiian bobtail squid, *Euprymna scolopes*. The partnership between the two organisms works on many levels. Easily observable is the effect where the luminescence produced by bacterial cells colonizing parts of the squid body creates a kind of camouflage in a dark environment, which reduces the risk of attack by squid predators [39]. Another well-known example in animal systems is that the gut microbiota communicate with their hosts through the production of short-chain fatty acids (SCFAs), such as acetate, propionate, and butyrate (see Figure 3). These SCFAs act as signaling molecules that influence various physiological processes in the host. For instance, SCFAs modulate immune responses by interacting with G-protein-coupled receptors on immune cells, enhancing the production of anti-inflammatory cytokines and promoting regulatory T cell differentiation. This helps maintain gut homeostasis and prevents excessive inflammatory responses. Beyond the immune system, microbial metabolites can alter the permeability of the intestinal lining, indirectly affecting brain function by modulating systemic inflammation and neural signaling. These complex interactions highlight the profound influence of gut microbiota-derived chemical signals on host health, immunity, and behavior [40].

### 2.4. Chemical Communication and Cell Differentiation of Microorganisms

Cellular differentiation in microorganisms can be classified as induced, triggered by signals from the external environment, or as a process that is an integral part of the cell cycle [41]. Chemical signaling molecules are important elements of the phenotypic differentiation of microorganisms. Physiological differentiation towards biofilm formation has been intensively studied for many years, and a number of structurally distinct molecules have been found that act as signaling molecules. Autoinducers typical for individual bacterial groups are described, such as N-acyl-homoserine lactones (AI-1) in G^−^ bacteria [42], signal peptides (AIP) in G^+^ bacteria [43], and furanosyl borate diester (AI-2) [44], which plays the role of a universal language between G^+^ and G^−^ bacteria. AI-1 and its key role in the quorum sensing mechanism was first described in the marine bacterium Vibrio fischeri, where it was involved in the expression of luminescence. Subsequently, AI-2 was also identified in this bacterium, which also interferes with the formation of luminescence [45,46]. AI-2 is involved in the morphological dedifferentiation of Lactobacillus plantarum, influencing the development of its biofilm phenotype and the amount of extracellular polysaccharides produced [47]. Other molecules that are often used as signaling molecules in bacterial differentiation, including biofilm formation and virulence, are the so-called diffusible signaling factor (DSF) or cis-2-unsaturated fatty acids. They have been found, for example, in the genera *Achromobacter*, *Yersinia*, *Serratia*, *Enterobacter*, and *Cronobacter* [48].

Romero et al. [21] also describe some antibiotics as signaling molecules. Antibiotics occur at sub-inhibitory concentrations in the natural environment and, under these conditions, can act as signaling molecules toward the microorganisms present. Regulatory effects on changes in cell morphology, transition to a biofilm phenotype, or influence on virulence factor production have been described [49]. In yeast, morphological differentiation is regulated by chemical signals, specifically the prenylated peptide pheromone a-factor and the unprenylated α-factor, each secreted by the opposite mating type. The differing properties of these pheromones are essential for communication and mating between yeast cells. For example, depending on nutrient availability, pseudohyphal growth can be induced in *Saccharomyces cerevisiae* [50,51]. Signaling molecules of filamentous fungi often have multiple interdependent roles. Beyond inducing morphological differentiation, they stimulate the production of various biologically active compounds, including pigments and toxins, which can also provide protective benefits to the producing organism (see Figure 4) [52]. Zearalenone is an estrogenic mycotoxin produced by Fusarium roseum, known to cause symptoms such as food refusal, diarrhea, and alimentary hemorrhage. Besides these effects, zearalenone plays a role in the formation of perithecia (fruiting structures) in the producing fungus itself and, through conditional gene expression, also impacts the physiology of the ascomycete *Gibberella zeae* (anamorph—*Fusarium graminearum*) [53].

The pigment melanin is often directly related to the development of fungal structures, increases the resistance of spores to UV radiation, and is also classified as a virulence factor in pathogenic fungi [52]. Melanin is also involved in the regulation of appressoria formation in many fungi. For example, in *Colletotrichum gloeosporioides*, non-melanin-forming mutants do not form appressoria, and the fungus is not pathogenic [54]. Microorganisms are capable of producing chemical compounds that are involved in the chemical differentiation of other, higher organisms. Secondary metabolites of streptomycetes can induce fruiting body formation in basidiomycetes. Examples of such compounds are anthranilate or basidifferquinone. The latter can induce fruiting body formation in *Favolous arcularius* in the absence of photo-irradiation, which is otherwise required for fruiting body formation in this basidiomycete [55].

Conidiogenone is a diterpene and was first described as a conidia-inducing compound in *Penicillium cyclopium*. It was later shown to be present in more than 10 species of *Penicillium*, confirming that it is a common signaling compound inducing conidiation in *Penicillium*. In addition to this activity, activity against HL-60 leukemia cells and also against, for example, MRSA, has also been described [56,57]. Some secondary metabolites have directly demonstrated the ability to differentiate animal cells. This finding is of disciplinary importance in the context of cancer cells. One example is rapamycin, a 31-membered macrocyclic natural product produced by the bacterium *Streptomyces hygroscopicus* [58]. It has been shown to induce the differentiation of human myeloid leukemia ML-1 cells [59].

## 3. Chemical Languages of Plants and Animals

Chemical communication between other organisms in this review is mainly focused on communication between microorganisms and plants or animals. Chemical communication between plants and animals will only be mentioned briefly, as there are many comprehensive reviews on these topics [60,61,62,63,64]. 

Plants often use volatile compounds (VOCs) for chemical communication [62]. The emitted volatiles can be perceived by the plant itself, but also by other plants in the vicinity. Receptors that allow the perception of VOCs are mainly located in plasma membranes, however, the exact mechanisms of the “perception” of the volatile signals are not explained in detail [65]. The VOCs involved in the chemical signaling of plants can be divided into several groups: terpenic substances, derivatives of fatty acids, phenylpropanoids or benzenoids [66].

Apart from VOCs, the important role of signaling lipids has recently been uncovered. Plasma membrane lipids (such as sphingolipids and glycerophospholipids) serve as mediators in microbial recognition and defense response molecules. Furthermore, lipid-derived compounds (e.g., jasmonic acid and phosphatidic acid) act as modulators of immune response and help establish beneficial symbiosis [67]. In addition to communication based on chemical compounds, systems based on electrical signals or mechanical stimulation have been described in plants [68,69].

Communication based on chemical signaling also takes place in the root system of plants [70]. This is especially important under stress conditions, when plants produce organic acids and phenolic compounds which can influence both root microbiome composition and nutrient acquisition. For example, during periods of iron deficiency the release of scopoletin can enhance iron uptake and selectively promote the presence of beneficial microorganisms. Benzoxazinoids and terpenoids also serve dual roles in this context, providing direct defense against biotic and abiotic stress while supporting growth-promoting microbes [71,72]. 

The usual forms of animal communication are mechanical and verbal. But even here, communication is of a chemical nature and the substances involved are pheromones. These secreted substances elicit different types of responses from members of the same species. Pheromones play critical roles in various behaviors, such as mating, territorial marking, alarm signaling, and social organization in species like insects, mammals, and fish. These chemical signals are highly diverse in structure (e.g., hydrocarbons, esters and alcohols, steroids, and peptides), ranging from simple volatile molecules to complex peptides, and are often tailored to the specific needs of the species [63,64]. 

However, communication between animals is not carried out solely through products of their own metabolism. Research has confirmed that the metabolites of members of their individual microbiota are also used for chemical signaling between animals. These play a role in health status detection, kin recognition and group-specific signaling [73]. Beyond intraspecies interactions, animals also engage in chemical communication with other organisms, such as predators, prey, and symbiotic partners, highlighting the intricate ecological networks facilitated by chemical signals.

From an evolutionary point of view, the structure of chemical signals was influenced by environmental factors such as temperature, humidity, etc. These molecules had to have sufficient stability in the external environment to be effective. Given the similarity of vertebrate and invertebrate pheromones, parts of the chemical communication system seem to have undergone an evolutionary conservation [63,74].

## 4. Conclusions

The exploration of chemical signaling across diverse organisms highlights the intricate and varied forms of communication that support both cooperative and antagonistic interactions. Through mechanisms such as quorum sensing, organisms regulate gene expression and adapt population behavior, demonstrating a fundamental, interspecies language that influences microbial communities, plant–microbe interactions, and host–pathogen dynamics.

Additionally, the review highlights the dual functions of chemical signals, which can support defense mechanisms or facilitate symbiosis, contributing to ecological balance. Examining these complex signaling pathways provides insights relevant to applications in medicine, agriculture, and environmental management, where influencing microbial behavior holds potential.

Current instrumental analysis provides tools which can detect new chemical compounds in increasingly complex biological matrixes and at trace levels, and molecular biology techniques can elucidate their functions, often revealing signaling/communication functions in living organisms. In particular, the relevance of chemical communication in stress responses, pathogen dynamics and symbiotic relationships can help solve challenges in healthcare, for example overcoming antibiotic resistance or better management of system interactions in the body. Broadening the knowledge of signaling in shifting environments could help uncover new strategies for conservation and sustainable policies.

## Figures and Tables

**Figure 1 molecules-30-00431-f001:**
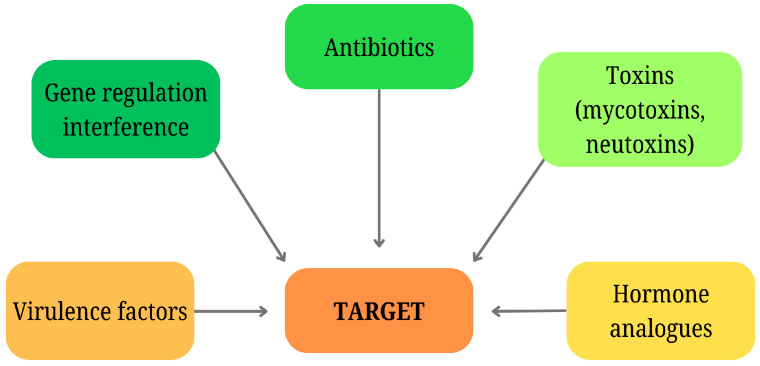
Different routes of microbial competition and predation through chemical signaling, demonstrating the production of toxic metabolites that influence resource access, predator deterrence, and interspecies dynamics.

**Figure 2 molecules-30-00431-f002:**
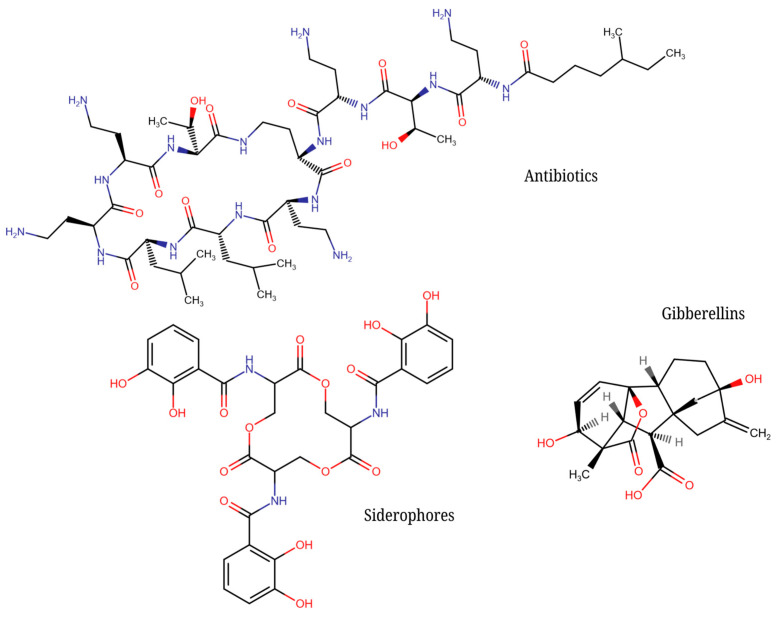
Examples of chemical compounds produced by microorganisms, categorized into antibiotics (colistin), siderophores (enterobactin), and plant hormones (gibberellic acid), illustrating their chemical diversity.

**Figure 3 molecules-30-00431-f003:**
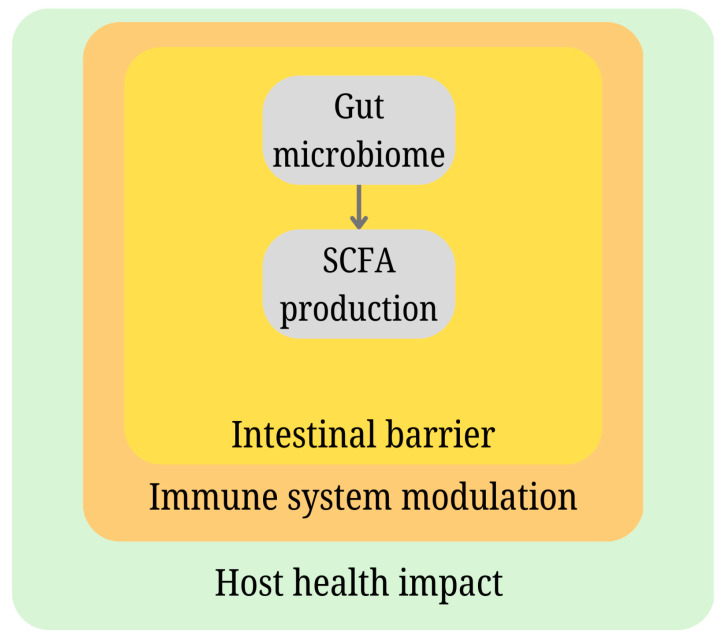
Representation of chemical communication between gut microbiota and host organisms, focusing on the role of microbial metabolites such as short-chain fatty acids (SCFAs) in immune modulation and gut–brain axis regulation.

**Figure 4 molecules-30-00431-f004:**
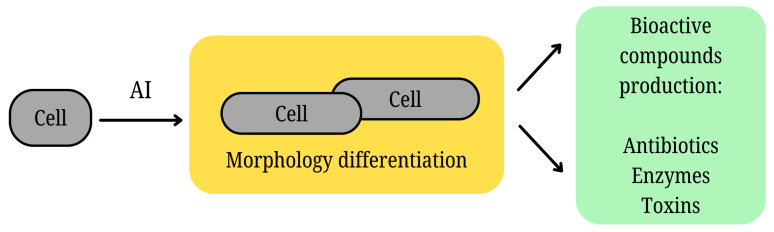
Chemical signaling and differentiation in microorganisms, showcasing the role of signaling molecules in morphological changes, secondary metabolite production, and microbial interactions.

**Table 1 molecules-30-00431-t001:** Examples of signaling molecules produced by microorganisms, their sources, and primary functions in chemical communication.

Signaling Molecule	Producer Organism	Key Functions
Acyl-homoserine lactones (AHLs)	Gram-negative bacteria (e.g., *Pseudomonas aeruginosa*)	Quorum sensing in Gram-negative bacteria
γ-Butyrolactones	*Streptomyces* sp.	Regulation of secondary metabolite production
Autoinducing peptides (AIPs)	Gram-positive bacteria (e.g., *Staphylococcus aureus*)	Quorum sensing in Gram-positive bacteria
AI-2 (furanosyl borate diester)	Both Gram-negative and Gram-positive bacteria	Universal quorum sensing molecule
Competence-stimulating peptide (CSP)	Various bacteria (e.g., *Streptococcus*, *Bacillus*)	Regulation of competence
Nisin	*Lactococcus* sp.	Antibacterial role
c-di-GMP	Various bacteria	Second messenger
Cyclic di-AMP (c-di-AMP)	Various bacteria	Second messenger
Cyclic AMP (cAMP)	Various bacteria	Second messenger
Diffusible signaling factor (DSF)	Gram-negative bacteria (e.g., *Xanthomonas*)	Biofilm formation, virulence
Pseudomonas quinolone signal (PQS)	*Pseudomonas aeruginosa*	Regulation of quorum sensing
Pyocyanin	*Pseudomonas aeruginosa*	Redox signaling
Indole	Various bacteria	Intercellular signal molecule, biofilm formation, antibiotic resistance
Farnesol	*Candida albicans*	Inhibits filamentous growth and regulates morphogenesis
Tyrosol	*Candida albicans*	Promotes yeast-to-hyphal transition and biofilm formation
Farnesoic acid	Various fungi	Hyphal growth inhibition
Indole-3-acetic acid (IAA)	Various fungi	Plant hormone mimicry, promotes plant–microbe interactions
1-Phenylethanol	*Saccharomyces cerevisiae*	Biofilm formation
Oxylipins	Various fungi	Regulation of fungal reproduction and signaling
Pheromones: a-factor, α-factor	*Saccharomyces cerevisiae*	Mating type coordination
Volatile compounds	*Candida albicans*	Filamentation regulation
Tryptophol	*Candida albicans*	Autoantibiotic action, filamentation inhibition

## Data Availability

No new data were created or analyzed in this study. Data sharing is not applicable.

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
