# Peer review of "Chemical Conversations"

_molecules, 2025, doi:10.3390/molecules30030431_

Round 1
Reviewer 1 Report
Comments and Suggestions for Authors
This review article is a comprehensive and insightful contribution to the understanding of chemical communication in microorganisms, plants, and animals. It successfully illustrates the diversity of chemical signaling systems and their ecological importance. The article would benefit from a dedicated section discussing future research directions. Given the complexity and potential applications of chemical communication, it would be valuable to highlight areas where further research is needed, such as the role of chemical signals in climate change, disease control, and ecosystem restoration.
Author Response
Dear Reviewer,
Thank you for your careful reading of the manuscript. Your recommendations for further research and possible applications of chemical communication are in the final chapter.
Reviewer 2 Report
Comments and Suggestions for Authors
The manuscript, entitled Chemical Conversations, is part of the Chemical Biology section of the journal Molecules. This review provides examples of chemical communication between microorganisms, between microorganisms and plants, and between microorganisms and animals.
The structure of the article is correct. It consists of a textual and graphical abstract and chapters: Introduction, Chemical languages of microorganisms, Chemical languages of plants and animals, Conclusion and References.
All the chapters are well written. They are supported by examples and the conclusions follow from the literature review. The authors of the manuscript document and systematise the state of the art of chemical communication between organisms.
The only flaw in the manuscript is the sloppy compilation of the bibliography. Some entries contain the full names of journals, while others contain abbreviated names. The list should also be preceded by the word References. One may also wonder about the advisability of including a graphical abstract in the proposed form.
Author Response
Dear Reviewer,
Thank you for your careful study of the manuscript and for pointing out the errors in the bibliography, for which we apologise. I hope everything has been corrected in the new version of the manuscript. The graphic summary could be more graphically perfect, but we believe it illustrates the content of this review quite well.
Reviewer 3 Report
Comments and Suggestions for Authors
The present work presents an interesting and complete compilation of microbial capabilities related to the synthesis of molecules involved in intra- and interspecies communication events. The paper is well written and reads easily. As a less favorable aspect, perhaps it would be convenient to expand the section dedicated to the relationships established with plants and animals, since the imbalance with respect to the part dedicated to events between microorganisms is important.
Some typographical errors:
L. 72: delete point after tyrosol.
L.72: Tab. I (replace by Tab. 1).
L.171: Fusarium in italics.
L.177-178: delete italics in spp.
L. 209: delete period after repellent.
L. 225-226: wording “Other well known example in animal systems, it is well known that gut...”
L. 252: Vibrio fischri.
L. 302: delete italics in bibliographic citation.
L. 313: delete period after detail.
Author Response
Dear Reviewer,
Thank you for your careful study of the manuscript and for pointing out some errors. We have corrected them in the new version of the manuscript. We also agree that the chapter on chemical signalling in plants and animals is brief. This topic is very broad and to be comprehensive enough would require a significant expansion of the manuscript. In order to keep the text compact, we have chosen this brief, informative form, supplemented by appropriate citations.
Reviewer 4 Report
Comments and Suggestions for Authors
I have carefully reviewed the manuscript entitled “Chemical Conversations” submitted by Michailidu et al. to a MDPI journal Molecules. This review presents examples of chemical communication among microorganisms, between microorganisms and plants, and between microorganisms and animals and meanwhile, the mechanisms and physiological importance of this communication are described here. Chemical interactions can be both cooperative and antagonistic. Microbial chemical signals usually ensure the formation of the most advantageous population phenotype or the disadvantage of a competitive species in the environment. The authors also described symbiotic (e.g., in the root system) and parasitic relationships between microorganisms and plants. Similarly, mutually beneficial relationships are established between microorganisms and animals (e.g., gastrointestinal tract), but microorganisms also invade and disrupt the immune and nervous systems of animals. I must admit that the manuscript offers the valuable insights into the interactions between microorganisms and plants and animals and can provide some guidance for my personal work in the futher. However, due to the limited length, I feel that there are many important issues that have not been fully described. The chemical communications between organisms are a very grand proposition, and it is currently at the forefront of research in the field of biology, especially microbiology. However, this manuscript only includes 69 references, which is obviously very insufficient. The topic of this manuscript focused more on the microorganisms, and therefore, the author should concentrate on a specific direction and narrow down the scope mentioned in the review, so that it is easier to write valuable review article in a certain aspect, for example, focusing solely on the interaction between microorganisms and plants or solely on the interaction between microorganisms and animals. Some minor suggestions are as the following:
1. Adding a new section to describe main chemical signal substances;
2. Soil nutrient concentrations are commonly a type of chemical signal, and therefore, it should add a section to describe the reaction of rhizosphere/root zone/soil microorganisms and plants.
3. Adding a new section to describe commercial chemical substances in actual agricultural production and their effectiveness.
Author Response
Dear Reviewer,
Thank you very much for your careful study of the manuscript and for the enclosed suggestions with which we fully agree. However, in the 2 days we have to correct the text, it would be very difficult to make the suggested changes in an adequate quality. You are absolutely right that the number of citations could be multiplied many times over. We have tried to limit ourselves to high quality publications where there is comprehensive information on the topic cited. Concerning the focus of the publication, we chose to compromise between a much narrower specialisation of a certain area of this broad topic (such information can often be found in the literature cited) and preparing a much longer text that would be more difficult for the reader to grasp.